# Comprehensive Evaluation of Inflammatory Biomarkers in Cervical Cancer Treated with Chemoradiotherapy

**DOI:** 10.3390/curroncol32010039

**Published:** 2025-01-13

**Authors:** Timur Koca, Nurcihan Gocen Vardar, Rahmi Atıl Aksoy, Aylin Fidan Korcum

**Affiliations:** 1Department of Radiation Oncology, Akdeniz University, 07070 Antalya, Turkey; nurcihan@akdeniz.edu.tr (N.G.V.); aylinkorcum@akdeniz.edu.tr (A.F.K.); 2Department of Radiation Oncology, Izmir City Hospital, 35540 Izmir, Turkey; rahmiatil.aksoy@saglik.gov.tr

**Keywords:** cervical cancer, inflammation, biomarkers, chemotherapy, radiotherapy, survival

## Abstract

**Objective**: Inflammatory biomarkers have been shown to possess both prognostic and predictive significance in various cancers. Among the emerging biomarkers, the pan-immune-inflammation value (PIV) has recently been introduced as a novel indicator representing both the immune response and the systemic inflammatory state. This study aims to comprehensively evaluate the predictive value of inflammatory biomarkers on survival outcomes in cervical cancer patients undergoing chemoradiotherapy. **Methods**: A total of 90 patients who had undergone chemoradiotherapy for cervical cancer were included. Data on demographics, treatment protocols, pre-treatment blood parameters, and survival outcomes were collected. The association between inflammatory biomarkers and survival outcomes was investigated through univariate and multivariate analyses. **Results:** The univariate analysis identified the following as predictors of progression-free survival (PFS): neutrophil–lymphocyte ratio (NLR), platelet–lymphocyte ratio (PLR), monocyte–lymphocyte ratio (MLR), systemic immune-inflammation index (SII), PIV, C-reactive protein (CRP), albumin, and tumor size. Multivariate analysis revealed that only the PIV significantly predicted PFS (HR 3.05, 95% CI 1.0 to 9.3, *p* = 0.04). In the univariate analysis, several variables were predictive of overall survival (OS), including NLR, PLR, MLR, SII, PIV, CRP, LDH, albumin, tumor size, and Eastern Cooperative Oncology Group Performance Status (ECOG PS). Multivariate analysis revealed CRP (HR 3.41, 95% CI 1.5 to 7.7, *p* = 0.003) and ECOG PS (HR 4.78, 95% CI 1.3 to 17.3, *p* = 0.01) predictive of OS, with PIV approaching statistical significance (HR 2.56, 95% CI 0.8 to 7.6, *p* = 0.09). **Conclusions**: This study provides the first comprehensive analysis of the association between cervical cancer and various inflammatory biomarkers. Many of these biomarkers have demonstrated predictive value for survival outcomes in patients with cervical cancer undergoing definitive chemoradiotherapy. Among the biomarkers evaluated, CRP and PIV were identified as the most predictive, warranting further exploration in future research.

## 1. Introduction

Cervical cancer represents the fourth leading cause of cancer-related mortality among women, with evidence suggesting an elevated risk of disease development among young women in certain regions [1]. Among women aged 20–39, cervical cancer remains the second leading cause of cancer-related deaths worldwide [2]. The 1- and 5-year overall acsurvival (OS) rates for 2004–2009 were 82.9% and 62.8%, respectively [3]. Although clinicopathological factors such as histopathological type, tumor size, tumor grade, depth of invasion, lymph node involvement, and lymphovascular invasion have been recognized as prognostic and predictive markers in cervical cancer, the identification and validation of novel factors may further enhance prognostication and improve treatment outcomes [4,5].

Recent studies have highlighted the critical role of inflammation in the initiation and progression of various solid malignancies [6,7]. The prognostic value of inflammatory biomarkers, including the neutrophil-to-lymphocyte ratio (NLR), monocyte-to-lymphocyte ratio (MLR), platelet-to-lymphocyte ratio (PLR), eosinophil-to-lymphocyte ratio (ELR), and fibrinogen-to-albumin ratio (FAR), has been extensively investigated in cervical cancer [8,9,10,11,12]. However, the limited discriminative capacity of these single biomarkers constrains their clinical applicability. Given the complex interactions among immunity, inflammation, and cancer, it is hypothesized that composite biomarkers, which capture the overall inflammatory environment, may offer more stable and robust predictive capabilities. Notably, the systemic inflammation response index (SIRI) and systemic immune-inflammation index (SII) have been employed to stratify the prognostic outcomes of patients with cervical cancer [13,14]. Additionally, the pan-immune-inflammation value (PIV), a recently characterized biomarker that encompasses multiple components of peripheral blood immune cell subsets such as neutrophils, monocytes, platelets, and lymphocytes, offers the potential to comprehensively represent both the immune response and systemic inflammatory status in patients [15]. Previous studies and systematic reviews have identified the PIV as a robust prognostic marker for outcomes in various cancers [16,17,18]. Furthermore, the prognostic significance of PIV was highlighted in a recent study involving 847 patients with cervical cancer [19]. The objective of this study was to comprehensively assess the predictive value of inflammatory biomarkers on survival outcomes in cervical cancer patients receiving definitive chemoradiotherapy.

## 2. Materials and Methods

### 2.1. Patient Selection

Patients with cervical cancer who underwent curative chemoradiotherapy at our institution between January 2014 and April 2023 were included in this retrospective study. Patients aged 18 years and older, with a histopathologically confirmed diagnosis of cervical cancer, staged as IA-IVA according to the 2018 FIGO classification, and with serum laboratory results acquired within one month prior to the initiation of treatment, were included in the study. Patients with severe comorbidities resulting in an Eastern Cooperative Oncology Group performance status (ECOG PS) of 3–4, distant metastases, incomplete medical records, prior primary surgery, or non-curative treatment were excluded. In addition, conditions such as infections and chronic active inflammatory diseases, known to induce either an acute or chronic systemic inflammatory response, were excluded. The REMARK guidelines were followed in this retrospective study [20].

### 2.2. Treatment Details and Follow-Up Procedures

No modifications were made to patient treatment or follow-up protocols for this study. The clinical procedures followed are outlined below. All patients underwent external beam radiotherapy (EBRT) with doses ranging from 45 to 50 Gy, administered in fractions of 1.8 to 2 Gy, with or without an additional boost up to 54 to 60 Gy. The EBRT techniques were intensity-modulated radiotherapy (IMRT), three-dimensional conformal radiotherapy (3D-CRT), and helical tomotherapy. All patients underwent concurrent definitive radiotherapy with platinum-based chemotherapy. Following primary treatment, 80 patients (88.9%) received brachytherapy. Patients were followed up every 3 months during the first 2 years and every 6 months during the subsequent 3 years following the completion of treatment. Patients were followed up using physical examination, positron emission tomography/computed tomography (PET/CT), and magnetic resonance imaging (MRI).

### 2.3. Data Collection

The data were collected from hospital records, as the studied predictive factors included age, ECOG PS, histopathology, tumor stage, the primary tumor’s maximum standardized uptake value (SUVmax), and inflammatory biomarkers. Patients were staged with MRI and PET/CT at the time of diagnosis using NCCN guidelines version 1.2024. All included patients have been assigned FIGO staging according to the 2018 FIGO staging system. Blood samples were taken within a month before the radiotherapy treatment to measure biomarkers of inflammation. The most recent results were used if the patient had multiple blood tests within one month of treatment.

Biochemical data of C-reactive protein (CRP) (mg/L), albumin (g/dL), and LDH (U/L) were obtained. The MLR was computed as monocyte count (10^3^/mL)/lymphocyte count (10^3^/mL), the NLR as neutrophil count (10^3^/mL)/lymphocyte count (10^3^/mL), the PLR as platelet count (10^3^/mL)/lymphocyte count (10^3^/mL).

The SII was derived using the formula: neutrophil count (10^3^/mL) × platelet count (10^3^/mL)/lymphocyte count (10^3^/mL). Similarly, the PIV was determined as neutrophil count (10^3^/mL) × platelet count (10^3^/mL) × monocyte count (10^3^/mL)/lymphocyte count (10^3^/mL) [15,21]. Hemo-eosinophil inflammation index (HEI) was calculated using the following criteria: hemoglobin <12 g/dL, SII > 560 and eosinophil count ≥100/μL. Those with a score of 0 to 1 were considered low risk, and those with a score of 2 to 3 were considered high risk [22].

### 2.4. Statistical Analyses

Statistical analysis was conducted using IBM SPSS version 27.0 software (IBM Corp. 2016, Armonk, NY, USA), with a significance level set at *p* < 0.05. Descriptive statistics were employed to characterize the patient population. The cut-off values for the variables were defined using receiver operating characteristic (ROC) curve analysis based on the Youden Index.

The primary outcomes analyzed in our study were OS, progression-free survival (PFS), and local recurrence-free survival (LRFS). OS was determined as the interval from the cervical cancer diagnosis to either death from any cause or the most recent follow-up assessment. PFS is the duration from diagnosis to documented disease progression or death from any cause without progression. LRFS was defined as the time from diagnosis to the occurrence of local recurrence or death from any cause in the absence of local recurrence. For patients who were alive without evidence of progression or local recurrence, PFS and LRFS were recorded following confirmation of the absence of these events.

Survival curves were generated using the Kaplan–Meier method, and the log-rank test was applied to assess the statistical differences between these curves. Univariate Cox proportional hazards regression analyses were utilized to estimate the hazard ratio (HR) for PFS, LRFS, and OS. Following univariate analyses, variables demonstrating statistical significance (*p* < 0.05) were incorporated into the multivariate analysis. The hazard ratios (HR) along with 95% confidence intervals (CI) were calculated to estimate the risk associated with each biomarker.

## 3. Results

The characteristics of the patients are presented in Table 1. Most patients (65.6%) were 63 years of age or younger. At the time of diagnosis, 6.6% of patients were classified as Stage I–IIa, 27.8% as Stage IIb, 26.7% as Stage III, and 38.9% as Stage IV. In the pathological subtyping of cancer, the results of 76 patients (84.4%) were reported as squamous cell carcinoma. A total of 42 patients (46.7%) were identified as being associated with human papillomavirus (HPV), while the pathology results for 7 patients (7.7%) were found to be negative for HPV. The HPV status of the remaining 41 patients (45.6%) could not be determined. The ECOG-PS score was 0–1 in 85 patients (94.4%), while five patients (5.6%) scored 2. Fourteen patients (15.6%) underwent pelvic and paraaortic lymph node dissection, while the remaining 76 (84.4%) did not.

All patients received chemoradiotherapy, with 82 patients (91.2%) administered cisplatin and the remaining eight patients (8.8%) treated with carboplatin. A total of 14 patients (15.6%) underwent 3D-CRT, 47 patients (52.2%) received IMRT, and 29 patients (32.2%) were treated with helical tomotherapy. A total of 80 patients (88.9%) were treated with brachytherapy following EBRT. As demonstrated by the HEI value, 67 patients (74.4%) were classified as high risk, while 23 patients (25.6%) were classified as low risk. The median follow-up period for patients was 36.2 months (range 4.9–117.4 months).

The cut-off values were established through ROC curve analysis, as detailed in Table 2. The cut-off value for NLR was 4.14 with 42.9% sensitivity and 89.1% specificity; for PLR, it was 170.5 with 62.9% sensitivity and 74.5% specificity; for MLR, it was 0.30 with 65.7% sensitivity and 74.5% specificity; for SII, it was 1167 with 51.4% sensitivity and 83.6% specificity; for PIV, it was 635.1 with 62.9% sensitivity and 83.6% specificity; for CRP, it was 17.17 mg/L with 62.9% sensitivity and 87.3% specificity; for LDH, it was 242 U/L with 28.6% sensitivity and 92.7% specificity; for albumin, it was 4.18 g/dL with 62.9% sensitivity and 76.4% specificity; for SUV_max_ value of the tumor it was 17.65 with 51.7% sensitivity and 68.9% specificity and for tumor size it was 4.85 cm with 82.9% sensitivity and 60.4% specificity.

The PFS rates at one and two years were found to be 73.2% and 65.5%, respectively. The effect of clinical characteristics and inflammatory biomarkers on PFS is detailed in Table 3. In the univariate analysis, the following variables were identified as significant predictors of PFS: NLR (*p* = 0.01), PLR (*p* = 0.011), MLR (*p* = 0.001), SII (*p* = 0.008), PIV (*p* < 0.001), CRP (*p* = 0.021), albumin (*p* = 0.021), and tumor size (*p* = 0.004). The univariate analysis showed that the LDH, HEI, age, stage, ECOG PS, and SUV_max_ value did not significantly predict PFS. In the multivariate analysis, only PIV was identified as a statistically significant predictor of PFS (*p* = 0.04). The predictive value of the PIV and CRP for PFS is presented in Figure 1 and Figure 2, respectively.

The one-year and two-year LRFS rates were 85.3% and 76.7%, respectively. The impacts of clinical characteristics and inflammatory biomarkers on LRFS are illustrated in Table 4. The univariate analysis revealed that SII (*p* = 0.012), PIV (*p* = 0.003), albumin (*p* = 0.009), and tumor size (*p* = 0.036) were significantly predictive of LRFS. In the multivariate analysis, none of the variables were identified as statistically significant predictors of LRFS.

The one-year and two-year OS rates were 89.8% and 77.4%, respectively. The impact of clinical characteristics and inflammatory biomarkers on OS is presented in Table 5. The analysis revealed that multiple factors were strongly correlated with OS. Univariate analysis identified the following variables as significant predictors of OS: NLR (*p* < 0.001), PLR (*p* = 0.001), MLR (*p* < 0.001), SII (*p* < 0.001), PIV (*p* < 0.001), CRP (*p* < 0.001), LDH (*p* = 0.043), albumin (*p* = 0.001), tumor size (*p* = 0.001), and ECOG PS (*p* < 0.001). In the multivariate analysis, CRP (*p* = 0.003) and ECOG PS (*p* = 0.017) were statistically significant predictors of OS. Furthermore, the multivariate analysis indicated that PIV (*p* = 0.09) approached statistical significance for OS. The predictive significance of PIV and CRP for OS is shown in Figure 3 and Figure 4, respectively.

## 4. Discussion

Pathological factors such as tumor size, depth of stromal invasion, lymphovascular invasion, and lymph node involvement are identified as potential prognostic indicators in cervical cancer [23]. Additionally, the immune system plays a pivotal role in carcinogenesis, with immune cells from both the innate and adaptive branches infiltrating the tumor microenvironment and actively influencing tumor progression [24,25,26]. Inflammation is a critical aspect of the tumor microenvironment and significantly contributes to tumor initiation, progression, invasion, promotion, and metastasis [27]. Several factors can cause inflammation in cervical cancer. HPV infection is the most relevant causative factor in the development of cervical cancer [28,29]. Studies have also shown how the progression of endometrial carcinoma correlates with a response from the immune system [30,31].

The role of inflammatory biomarkers such as NLR, PLR, MLR, CRP, and PIV have been studied in the prognosis of cervical cancer [8,9,10,19]. This study represents the first comprehensive evaluation of these inflammatory biomarkers collectively. Numerous of these biomarkers were found to be significant predictors of survival in patients with cervical cancer undergoing chemoradiotherapy. Furthermore, multivariate analysis identified PIV as a critical predictor of PFS and CRP as a significant predictor of OS. These findings highlight the need for further prospective investigation into the predictive value of both biomarkers.

Neutrophils express several factors and cytokines that can influence the immune system and contribute to tumor growth and spread. These include angiogenesis factors such as matrix metalloproteinase 9 (MMP9) and vascular endothelial growth factor (VEGF), as well as cytokines such as transforming growth factor beta (TGF-β) and interferon beta (IFN-β). Recent studies have suggested that neutrophils play a role in tumor initiation and progression [32]. Higher NLR values have been associated with poor survival in patients with cervical cancer [33]. Additionally, Elmali and colleagues demonstrated that higher NLR values were an independent prognostic factor of poor LRFS and distant metastasis free survival (DMFS) [34]. Similarly, the findings of our study revealed a notable association between elevated NLR levels and shortened PFS and OS.

Platelets play a role in inflammatory processes; accordingly, the phenomenon of reactive thrombocytosis is more frequently observed in the context of solid tumors. Platelets can release several substances, such as VEGF, platelet-derived growth factor (PDGF), and TGF-β. These substances can stimulate the proliferation and adhesion of malignant tumor cells, thereby contributing to tumor growth and facilitating metastasis [35]. Thrombocytosis in patients with cervical cancer has been linked to a poor prognosis [36]. A study by Kawano and colleagues found that a higher platelet count was linked to more advanced clinical stages, larger tumor sizes, and an increased incidence of treatment failure [37]. A meta-analysis conducted by Templeton and colleagues on 12,754 patients indicated that an elevated PLR was associated with a significantly poorer OS outcome in solid tumors [38]. Additionally, another meta-analysis involving 3668 patients with cervical cancer identified a higher PLR as a significant predictor of poor prognosis [39]. Similarly, our study revealed that the PFS and OS were shorter in cervical cancer patients with higher PLR values.

CRP is a biomarker of inflammation synthesized in the liver during an acute inflammatory response. The primary biological function of this system is to identify pathogens and damaged cells within the host organism and to facilitate their removal by recruiting components of the immune system and phagocytic cells [40]. High pre-radiotherapy CRP levels have been correlated with poorer OS [41]. Furthermore, a meta-analysis including 2204 patients with cervical cancer determined a higher CRP as a significant predictor of poor PFS and OS [42]. Similarly, the present study revealed a shorter PFS and OS in cervical cancer patients with elevated CRP values.

Previous studies have indicated that higher SII levels are associated with poorer LRFS, disease-free survival (DFS), and OS outcomes in patients with various cancers [43,44]. Additionally, Eraslan and colleagues demonstrated that SII is a predictor of pathological complete response in patients with locally advanced rectal cancer receiving neoadjuvant chemoradiotherapy [45]. Huang and colleagues reported that the SII can independently predict OS in patients with cervical cancer undergoing radical resection [13]. Additionally, a recently published study revealed that SII can predict OS in patients with cervical cancer receiving definitive chemoradiotherapy [46]. Moreover, our study showed that SII was a significant predictor of PFS, LRFS, and OS in patients with cervical cancer undergoing definitive chemoradiotherapy.

While the prognostic and predictive significance of several inflammatory biomarkers in solid tumors, particularly cervical cancer, has been established, there remains a compelling need for the development of new and more robust biomarkers to enhance outcome prediction. The PIV is an emerging and effective biomarker for predicting clinical outcomes in cancer patients. In a study conducted by Baba and colleagues on patients with esophageal cancer, those with lower PIV levels demonstrated a reduced overall mortality rate [47]. Fuca and colleagues observed that PIV exhibited a considerable and robust prognostic influence on both PFS and OS in patients with metastatic colorectal cancer undergoing first-line chemotherapy [15]. Our previous research demonstrated that higher PIV levels were associated with shorter survival outcomes in laryngeal cancer patients treated with radiotherapy [48]. Additionally, a recent study has shown that in patients with locally advanced cervical cancer, PIV is correlated with cancer progression and has poorer therapeutic outcomes, DFS, and OS [19]. Similarly, in this study, our findings indicated that elevated PIV levels are associated with reduced PFS, LRFS, and OS in cervical cancer patients treated with chemoradiotherapy. Additional research is required to evaluate the association of other inflammatory markers with progression in cervical cancer.

Our study findings indicated that the PFS and OS were shorter in cervical cancer patients with higher MLR values. Chen and colleagues found that a high MLR was significantly associated with more advanced tumor stage and parametrial involvement [49]. In another study, elevated MLR values were significantly linked to advanced stage and lymph node involvement [50]. Ronsini and colleagues found that diffuse lymphovascular invasion was linked with poorer DFS compared to focal or absent involvement in patients with cervical cancer [51]. Additional research is required to understand better the extent to which the levels of these markers are related to these specific mechanisms. In this context, future correlation of parameters such as lymphovascular invasion, stromal infiltration, and tumor size with inflammatory markers may contribute to a more comprehensive understanding of the role of inflammation in cancer.

The study’s results indicated that although the initial cut-off value of CRP was relatively high, 32.2% of patients remained above this cut-off. This finding suggests that the cut-off value does not apply to only a small minority of patients. Therefore, CRP may be considered a clinically useful prognostic marker. Similarly, the PIV, MLR, PLR, SII, and albumin cut-off values represent approximately one-third of the patient group. They can potentially be helpful in clinical practice as a general measure of systemic inflammation. These ratios support the view that these parameters may be statistically meaningful and in a clinical context. The high LDH cut-off value was statistically significant; however, its applicability to clinical practice may be limited. The high LDH cut-off value resulted in a smaller group of patients (15.6%) being included in this range, potentially affecting our results’ generalizability.

This study provides a comprehensive evaluation of the predictive role of inflammatory biomarkers in cervical cancer patients undergoing definitive chemoradiotherapy. However, it is essential to acknowledge the limitations of this study. The research employs a retrospective design, and the dataset is derived from a single center, encompassing a limited number of cases. Because our study was retrospective, some of our data did not provide p16 immunohistochemistry results. Also, HPV test results were not accessible for all patients. Future studies could further investigate the relationship between inflammatory markers and these parameters by evaluating p16 results and HPV infection status. Another potential limitation of this study is that the results of peripheral blood cell analysis may be susceptible to various confounding factors, such as infection, blood circulation capacity, and nutritional status. While patients with conditions known to affect parameters were excluded from the study, the circulating cell counts may still have been influenced by other unknown or undetectable factors. In the case of patients with more than one blood test result within one month before treatment, the result closest to the treatment date was used as the basis for the analysis. It is also possible that this method may not entirely exclude cases in which blood parameters are not stable, which represents another potential limitation of our study.

## 5. Conclusions

This study provides a novel and comprehensive examination of the relationship between cervical cancer and a range of inflammatory biomarkers. Numerous of these biomarkers have demonstrated predictive value for survival outcomes in patients with cervical cancer undergoing definitive chemoradiotherapy. Multivariate analysis indicated PIV as a significant predictor of PFS and CRP as a critical predictor of OS. The CRP and PIV may help better risk stratification for prognosis prediction of patients with cervical cancer. These findings may contribute to decision support systems in managing cervical cancer. If validated by larger studies, a simple pre-treatment blood sample may predict the survival of cervical cancer patients and help guide treatment strategies to determine the best approach for each patient.

## Figures and Tables

**Figure 1 curroncol-32-00039-f001:**
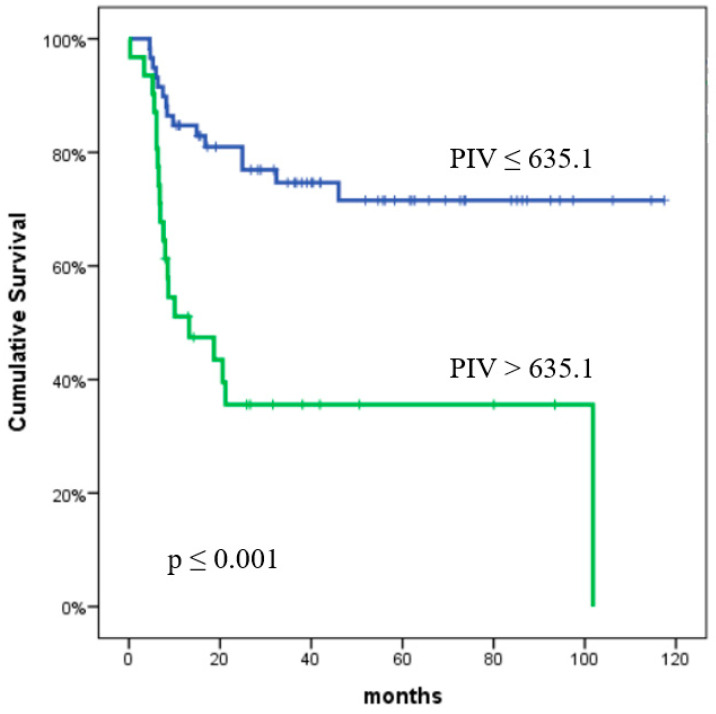
Progression-free survival according to the pan-immune-inflammation value. The *p* values were calculated using the log-rank test.

**Figure 2 curroncol-32-00039-f002:**
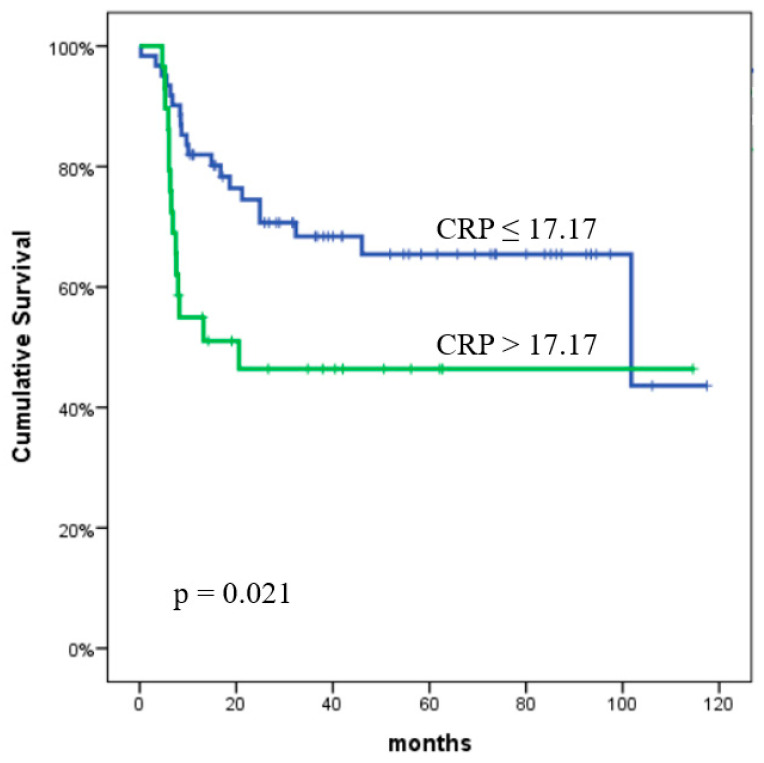
Progression-free survival according to the C-reactive protein value. The *p* values were calculated using the log-rank test.

**Figure 3 curroncol-32-00039-f003:**
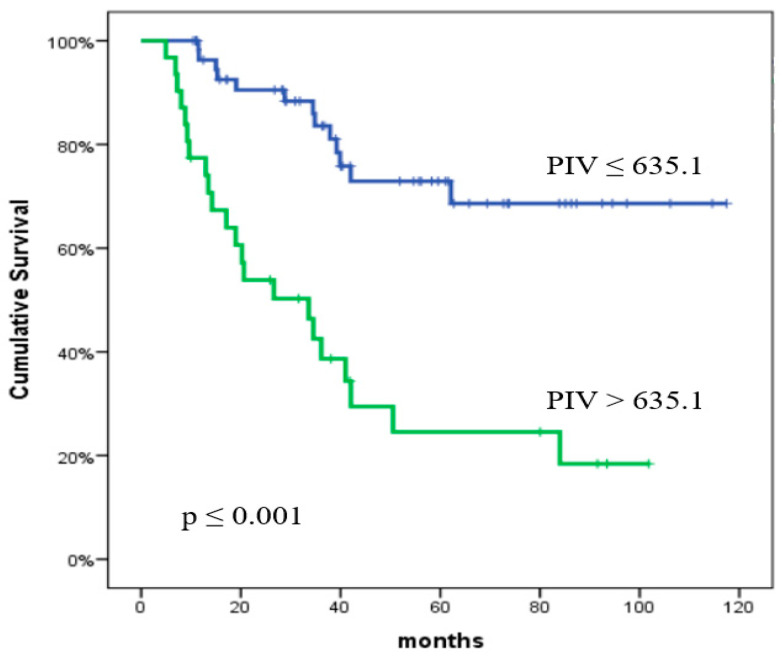
Overall survival according to the pan-immune inflammation value. The *p* values were calculated using the log-rank test.

**Figure 4 curroncol-32-00039-f004:**
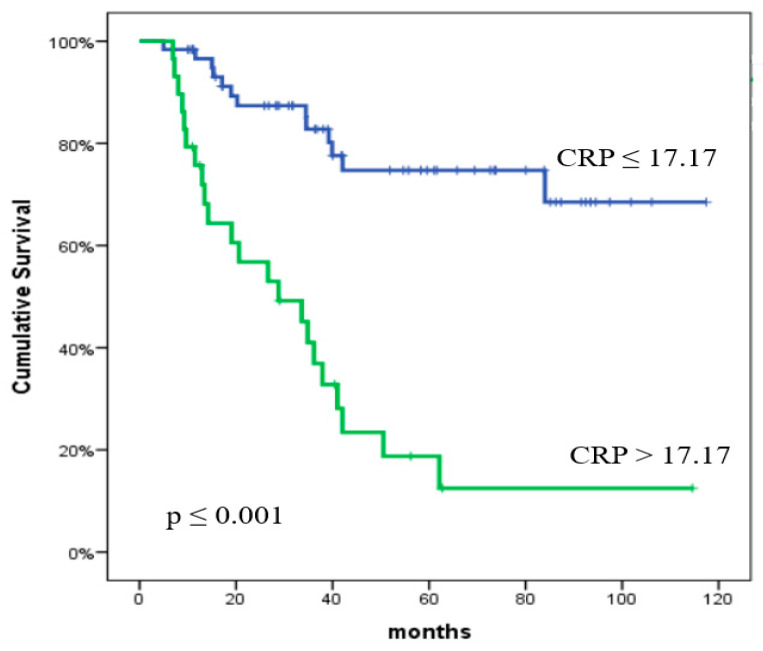
Overall survival according to the C-reactive protein value. The *p* values were calculated using the log-rank test.

**Table 1 curroncol-32-00039-t001:** Patient characteristics (n = 90).

Characteristic	n	%
Age (years)		
≤63	59	65.6
>63	31	34.4
Histopathology		
Squamous cell	76	84.4
Adenocarcinoma	7	7.8
Others	7	7.8
T Stage		
T1	2	2.2
T2	48	53.3
T3	4	4.5
T4	36	40
N Stage		
N0	53	58.9
N1	21	23.3
N2	10	11.1
N3	6	6.7
FIGO Stage		
Stage I–IIa	6	6.6
Stage IIb	25	27.8
Stage III	24	26.7
Stage IV	35	38.9
HPV		
Yes	42	46.7
No	7	7.7
Unknown	41	45.6
ECOG-PS		
0–1	85	94.4
2	5	5.6
PPLND		
Yes	14	15.6
No	76	84.4
Concurrent Chemotherapy		
Cisplatin	82	91.2
Carboplatin	8	8.8
Total EBRT Dose		
45 Gy	20	22.2
50 Gy	60	66.7
Other	10	11.1
Radiotherapy Modality		
3D-conformal radiotherapy	14	15.6
Intensity modulated radiotherapy	47	52.2
Helical tomotherapy	29	32.2
Brachytherapy		
Yes	80	88.9
No	10	11.1
Consolidation Chemotherapy		
Yes	4	4.4
No	86	95.6
Hemo-Eosinophil Inflammation Index		
High risk	67	74.4
Low risk	23	25.6
Tumor Size (cm)		
≤4.85	38	42.2
>4.85	50	55.6
Unknown	2	2.2
SUV_max_ of Tumor		
≤17.65	45	50
>17.65	29	32.2
Unknown	16	17.8

Abbreviations: HPV, human papillomavirus; ECOG PS, Eastern Cooperative Oncology Group—Performance Status; EBRT, external beam radiotherapy; PPLND, pelvic and para-aortic lymph node dissection.

**Table 2 curroncol-32-00039-t002:** ROC curve analysis for the prediction of overall survival.

Variable	Cut-Off	Sensitivity (%)	Specificity (%)	AUC (95% CI)	*p*	Patient Number Above the Cut-Off	Patient Number Below the Cut-Off
NLR	4.14	42.9	89.1	0.69 (0.57–0.80)	0.002	21 (23.3%)	69 (76.7%)
PLR	170.5	62.9	74.5	0.68 (0.56–0.80)	0.004	36 (40%)	54 (60%)
MLR	0.30	65.7	74.5	0.69 (0.57–0.80)	0.002	37 (41.1%)	53 (58.9%)
SII	1167	51.4	83.6	0.69 (0.57–0.81)	0.002	27 (30%)	63 (70%)
PIV	635.1	62.9	83.6	0.71 (0.60–0.83)	0.001	31 (34.4%)	59 (65.6%)
CRP	17.17	62.9	87.3	0.77 (0.66–0.87)	<0.001	29 (32.2%)	61 (67.8%)
LDH	242.0	28.6	92.7	0.55 (0.42–0.68)	0.413	14 (15.6%)	76 (84.4%)
Albumin	4.18	62.9	76.4	0.71 (0.59–0.83)	0.001	56 (62.2%)	34 (37.8%)
Age	63	45.7	72.7	0.56 (0.44–0.69)	0.284	31 (34.4%)	59 (65.6%)
Tumor Size	4.85	82.9	60.4	0.70 (0.59–0.81)	0.001	50 (55.6%)	38 (42.2%)
SUV_max_	17.65	51.7	68.9	0.59 (0.46–0.72)	0.18	29 (32.2%)	45 (50%)

Note: Tumor size is measured and expressed in centimeters. Abbreviations: AUC, area under curve; CI, confidence interval; NLR, neutrophil–lymphocyte ratio; PLR, platelet–lymphocyte ratio; MLR, monocyte–lymphocyte ratio; SII, systemic immune-inflammation index; PIV, pan-immune-inflammation value; CRP, C-reactive protein; LDH, lactate dehydrogenase.

**Table 3 curroncol-32-00039-t003:** Univariate and multivariate Cox regression analysis for the prediction of progression-free survival.

		Univariate Analysis		Multivariate Analysis	
Variable	Cut-Off	HR (%95 Cl)	*p*	HR (%95 Cl)	*p*
NLR	≤4.14 vs. >4.14	2.54 (1.25–5.18)	**0.010**	1.14 (0.36–3.61)	0.812
PLR	≤170.5 vs. >170.5	2.41 (1.22–4.76)	**0.011**	1.33 (0.54–3.25)	0.532
MLR	≤0.30 vs. >0.30	3.36 (1.68–6.73)	**0.001**	1.30 (0.47–3.56)	0.608
SII	≤1167 vs. >1167	2.51 (1.27–4.98)	**0.008**	0.53 (0.13–2.10)	0.373
PIV	≤635.1 vs. >635.1	3.73 (1.89–7.34)	**<0.001**	3.05 (1.00–9.30)	**0.049**
CRP	≤17.17 vs. >17.17	2.17 (1.10–4.25)	**0.021**	0.95 (0.40–2.25)	0.921
LDH	≤242 vs. >242	0.69 (0.24–1.96)	0.487		
Albumin	≤4.18 vs. >4.18	0.45 (0.23–0.88)	**0.021**	0.58 (0.22–1.50)	0.264
HEI	Low vs. High	1.95 (0.80–4.72)	0.137		
Age	≤63 vs. >63	1.06 (0.52–2.16)	0.854		
ECOG PS	0–1 vs. 2	1.68 (0.40–7.09)	0.476		
Stage	I-IIa vs. IIb-III-IV	23.4 (0.11–4652)	0.243		
Tumor size	≤4.85 vs. >4.85	3.19 (1.44–7.07)	**0.004**	1.96 (0.79–4.82)	0.143
SUV_max_	≤17.65 vs. >17.65	1.31 (0.61–2.81)	0.476		

Note: Tumor size is measured and expressed in centimeters. Abbreviations: HR, hazard ratio; CI, confidence interval; NLR, neutrophil–lymphocyte ratio; PLR, platelet–lymphocyte ratio; MLR, monocyte–lymphocyte ratio; SII, systemic immune-inflammation index; PIV, pan-immune-inflammation value, CRP, C-reactive protein; LDH, lactate dehydrogenase; HEI, Hemo-eosinophil inflammation index; ECOG PS, Eastern Cooperative Oncology Group—Performance Status.

**Table 4 curroncol-32-00039-t004:** Univariate and multivariate Cox regression analysis for the prediction of local recurrence-free survival.

		**Univariate Analysis**		**Multivariate Analysis**	
**Variable**	**Cut-Off**	**HR (%95 Cl)**	** *p* **	**HR (%95 Cl)**	** *p* **
NLR	≤4.14 vs. >4.14	2.23 (0.89–5.56)	0.085		
PLR	≤170.5 vs. >170.5	2.05 (0.86–4.86)	0.101		
MLR	≤0.30 vs. >0.30	2.30 (0.99–5.29)	0.050		
SII	≤1167 vs. >1167	3.02 (1.27–7.17)	**0.012**	0.75 (0.20–2.73)	0.666
PIV	≤635.1 vs. >635.1	3.57 (1.55–8.21)	**0.003**	2.93 (0.99–8.71)	0.052
CRP	≤17.17 vs. >17.17	0.94 (0.37–2.40)	0.902		
LDH	≤242 vs. >242	0.50 (0.11–2.17)	0.361		
Albumin	≤4.18 vs. >4.18	0.30 (0.12–0.74)	**0.009**	0.40 (0.13–1.18)	0.100
HEI	Low vs. High	1.78 (0.60–5.28)	0.296		
Age	≤63 vs. >63	1.20 (0.50–2.87)	0.674		
ECOG PS	0–1 vs. 2	1.39 (0.18–10.47)	0.748		
Stage	I-IIa vs. IIb-III-IV	23.7 (0.36–155)	0.339		
Tumor Size	≤4.85 vs. >4.85	2.71 (1.06–6.93)	**0.036**	1.73 (0.64–4.65)	0.275
SUV_max_	≤17.65 vs. >17.65	1.07 (0.41–2.76)	0.889		

Note: Tumor size is measured and expressed in centimeters. Abbreviations: HR, hazard ratio; CI, confidence interval; NLR, neutrophil–lymphocyte ratio; PLR, platelet–lymphocyte ratio; MLR, monocyte–lymphocyte ratio; SII, systemic immune-inflammation index; PIV, pan-immune-inflammation value, CRP, C-reactive protein; LDH, lactate dehydrogenase; HEI, Hemo-eosinophil inflammation index; ECOG PS, Eastern Cooperative Oncology Group—Performance Status.

**Table 5 curroncol-32-00039-t005:** Univariate and multivariate Cox regression analysis for the prediction of overall survival.

		Univariate Analysis		Multivariate Analysis	
Variable	Cut-Off	HR (%95 Cl)	*p*	HR (%95 Cl)	*p*
NLR	≤4.14 vs. >4.14	3.34 (1.70–6.56)	**<0.001**	1.36 (0.44–4.17)	0.590
PLR	≤170.5 vs. >170.5	3.19 (1.60–6.33)	**0.001**	1.44 (0.57–3.64)	0.438
MLR	≤0.30 vs. >0.30	3.83 (1.89–7.74)	**<0.001**	1.03 (0.40–2.63)	0.940
SII	≤1167 vs. >1167	3.31 (1.70–6.44)	**<0.001**	0.73 (0.18–2.99)	0.667
PIV	≤635.1 vs. >635.1	4.55 (2.28–9.06)	**<0.001**	2.56 (0.85–7.67)	0.093
CRP	≤17.17 vs. >17.17	5.54 (2.76–11.11)	**<0.001**	3.41 (1.51–7.73)	**0.003**
LDH	≤242 vs. >242	2.14 (1.02–4.46)	**0.043**	1.23 (0.51–2.98)	0.636
Albumin	≤4.18 vs. >4.18	0.32 (0.16–0.65)	**0.001**	0.81 (0.29–2.20)	0.682
HEI	Low vs. High	1.69 (0.70–4.10)	0.239		
Age	≤63 vs. >63	1.84 (0.94–3.59)	0.071		
ECOG PS	0–1 vs. 2	8.34 (2.74–25.32)	**<0.001**	4.78 (1.31–17.3)	**0.017**
Stage	I-IIa vs. IIb-III-IV	2.57 (0.35–18.80)	0.352		
Tumor Size	≤4.85 vs. >4.85	4.16 (1.72–10.02)	**0.001**	1.90 (0.63–5.68)	0.247
SUV_max_	≤17.65 vs. >17.65	1.56 (0.75–3.26)	0.232		

Note: Tumor size is measured and expressed in centimeters. Abbreviations: HR, hazard ratio; CI, confidence interval; NLR, neutrophil–lymphocyte ratio; PLR, platelet–lymphocyte ratio; MLR, monocyte–lymphocyte ratio; SII, systemic immune-inflammation index; PIV, pan-immune-inflammation value, CRP, C-reactive protein; LDH, lactate dehydrogenase; HEI, Hemo-eosinophil inflammation index; ECOG PS, Eastern Cooperative Oncology Group—Performance Status.

## Data Availability

The datasets generated and/or analyzed during the current study are not publicly available due to privacy but are available from the corresponding author upon reasonable request.

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
