# Peer review of "Comprehensive Evaluation of Inflammatory Biomarkers in Cervical Cancer Treated with Chemoradiotherapy"

_curroncol, 2025, doi:10.3390/curroncol32010039_

Round 1
Reviewer 1 Report
Comments and Suggestions for Authors
I read with great interest this Manuscript, which falls within the aim of the Journal.
Honestly, the topic is interesting enough to attract the readers’ attention. The methodology is accurate, and the data analysis supports conclusions. Nevertheless, authors should clarify some points and improve the discussion by citing relevant and novel critical articles.
- OVERALL COMMENTS:
- A native English speaker should further revise manuscript to improve clarity and readability.
- I want to inform You that I make a plagiarism check routinely, and I can confirm that Yours is an original writing.
- TITLE AND ABSTRACT:
- The title seems to me to be a simple but at the same time comprehensive title that makes the topic clear almost immediately.
- The abstract provides an informative and balanced summary of what was done and what was found.
- INTRODUCTION:
- Introduction is very clear, because it explains the scientific background and rationale of the manuscript is reported, furthermore it states explicit and specific objectives.
- METHODS:
- The study well describes the period of recruitment, the follow-up strategy and data collection.
- Inclusion criteria should be better clarified by extending their description, instead exclusion criteria are clearer.
- Methods could better define all outcomes, predictor factors and potential confounders, instead statistical methods are well explained.
- RESULTS AND DISCUSSION:
- Results report a number of outcome events or summary measures over time.
-Discussion of this study provides a very good general interpretation of the results.
- What are the actual clinical implications of this study? It is essential to report the results obtained by the authors in the context of clinical practice and to adequately highlight what contribution this study adds to the existing literature on the topic and to future study perspectives. Studies have shown how the progression of endometrial carcinoma correlates with a response from the immune system. Do you think the same may be true in cervical carcinoma? (Pls see PMID: 39537472; PMID: 39518008)
-What do you think are the factors that can lead to inflammatory reactions? Tumor growth? Stromal infiltration? LVSI? (Pls see PMID: 34116834)
Author Response
Thank you for your constructive and insightful comments.
We have made the following changes and additions based on your suggestions.
1-A native English speaker should further revise manuscript to improve clarity and readability.
The language of the manuscript has been revised and improved.
2- ‘Inclusion criteria should be better clarified by extending their description, instead exclusion criteria are clearer.’
We have added “The inclusion criteria are being aged ≥18, having a diagnosis of cervical cancer confirmed by histopathology, being staged as IA-IVA according to the 2018 FIGO staging system, and having serum laboratory results obtained from our hospital's blood analyzer within one month prior to treatment.” to the patient selection in materials and methods section.
3- Methods could better define all outcomes, predictor factors and potential confounders, instead statistical methods are well explained.
Dear peer reviewer, upon your suggestion, we have decided to provide further explanation about the results, predictor factors and potential confounding factors mentioned in the methodology section of our study.
We have added “The main outcomes analyzed in our study were OS, progression-free survival (PFS), and local recurrence-free survival (LRFS). OS was defined as the interval from the date of cervical cancer diagnosis to the date of death from any cause or the last follow-up. PFS was defined as the period from diagnosis to either documented disease progression or death from any cause in the absence of progression. LRFS was defined as the time from diagnosis to the occurrence of local recurrence or death from any cause in the absence of local recurrence. For patients who were alive without evidence of progression or local recurrence, PFS and LRFS were recorded following confirmation of the absence of these events.” to the method.
Additionally, we have added “The data was collected from hospital records, as the studied predictive factors included age, ECOG PS, histopathology, tumor stage, the maximum standardized uptake value (SUVmax) of the primary tumor, and inflammatory biomarkers.” to the method.
4- What are the actual clinical implications of this study? It is essential to report the results obtained by the authors in the context of clinical practice and to adequately highlight what contribution this study adds to the existing literature on the topic and to future study perspectives. Studies have shown how the progression of endometrial carcinoma correlates with a response from the immune system. Do you think the same may be true in cervical carcinoma? (Pls see PMID: 39537472; PMID: 39518008)
Thank you for your insightful feedback and for suggesting references to further contextualize our findings. Below, we address the clinical implications of our study and broaden the discussion and conclusion to include the potential parallels between immune responses in endometrial and cervical cancer.
We have added “PIV is correlated with cancer progression and had poorer therapeutic outcomes, DFS, and OS [43]. Further studies are needed to evaluate the association of other inflammatory markers with survival outcomes in cervical cancer.”
We have added “Studies have also shown how the progression of endometrial carcinoma correlates with a response from the immune system [51,52].”
We have added “Ronsini et al. analyzed that diffuse LVSI showed a worse disease-free survival (DFS) than patients with focal or absent involvement in cervical cancer [50]. Further studies are needed to better understand the extent to which the levels of these markers are related to these specific mechanisms. In this context, future correlation of parameters such as LVSI, stromal infiltration and tumor size with inflammatory markers may contribute to a more comprehensive understanding of the role of inflammation in cancer.” to the discussion.
Additionally, we have added “These findings may contribute to decision support systems in managing cervical cancer. If validated by larger studies, a simple pre-treatment blood sample may predict the survival of cervical cancer patients and help guide treatment strategies to determine the best approach for each patient.” to the conclusion.
5- What do you think are the factors that can lead to inflammatory reactions? Tumor growth? Stromal infiltration? LVSI? (Pls see PMID: 34116834)
Thank you for this insightful question and for directing us to the relevant literature. Tumor-induced inflammation in cervical cancer is a complex process influenced by several factors, including tumor growth, stromal infiltration and lymphovascular invasion. We have expanded the discussion in our manuscript to address these mechanisms.
We have added “Our study findings indicated that the PFS and OS were shorter in cervical cancer patients with higher MLR values. Chen et al. found that a high MLR was significantly associated with more advanced tumor stage and parametrial involvement [48]. In another study, a high MLR was significantly associated with advanced stage (IB3-IIIC2), lymphatic metastasis (total) and pelvic lymph node metastasis [49]. Ronsini et al analyzed that diffuse LVSI showed a worse disease-free survival (DFS) than patients with focal or absent involvement in cervical cancer [50].” to the discussion.
We have added “Further studies are needed to better understand the extent to which the levels of these markers are related to these specific mechanisms. In this context, future correlation of parameters such as LVSI, stromal infiltration and tumor size with inflammatory markers may contribute to a more comprehensive understanding of the role of inflammation in cancer.” to the discussion.
Reviewer 2 Report
Comments and Suggestions for Authors
Authors provide a detailed analysis of the association between cervical cancer and inflammatory biomarkers. In detail, CRP and PIV were identified as the most predictive biomarkers, warranting further exploration in future research.
Tha manuscript is well written and data are clearly presented. I have afew suggestions to improve the overall quality:
1) Immunohistochemistry for p16 should be documented in the studied cases since it represents a surrogate marker for HPV infection. Authors should specify p16 results especially in cases where HPV test was not available.
2) A brief discussion on the most relevant and emerging prognostic factors in cervical carcinoma should be included (please refer to PMID: 36831480)
Author Response
Thank you for your constructive and insightful comments.
We have made the following changes and additions based on your suggestions.
1- Immunohistochemistry for p16 should be documented in the studied cases since it represents a surrogate marker for HPV infection. Authors should specify p16 results especially in cases where HPV test was not available.
The use of p16 immunohistochemistry as a marker of HPV infection is very valuable in cervical cancer studies. However, because our study was retrospective, some of our data did not include p16 immunohistochemistry results. Also, HPV test results were not available for all patients.
Therefore, we could not directly assess HPV infection status in our study, and we have included this deficiency in the limitations section.
We have added “Because our study was retrospective, some of our data did not provide p16 immunohistochemistry results. Also, HPV test results were not accessible for all patients. Future studies could further investigate the relationship between inflammatory markers and these parameters by evaluating p16 results and HPV infection status.”
2- A brief discussion on the most relevant and emerging prognostic factors in cervical carcinoma should be included (please refer to PMID: 36831480)
We have reviewed the study you cited (PMID: 36831480) and found that it provides valuable insight into emerging prognostic factors in cervical cancer.
We have added “Several pathological parameters, including tumor size, depth of stromal invasion, lymphovascular invasion, and lymph node status, have been identified as potential prognostic indicators in cervical cancer [44].”
Reviewer 3 Report
Comments and Suggestions for Authors
This paper describes the usefulness of inflammatory biomarkers in predicting prognosis after CCRT for cervical cancer. Although there are many parameters, the author’s intentions and the flow of the paper are well-organized and easy to understand.
I have listed below the points that I found difficult to understand or that I thought should be clarified.
1. If possible, it may be better to indicate what PIV (pan-immune-inflammation value) is in the abstract. I think it is desirable that the abstract be self-contained.
2. In the M&M section, the authors stated that they excluded patients with infections and chronic inflammation. Were there any criteria for exclusion, and if not, who determined this? Because the cut-off value of CRP was high at 17.17, I suspect that patients with infection might be included.
3. Were there any cases where the patient had multiple blood test in the month before starting treatment? If so, which blood test results were accepted?
4. Was FIGO 2018? Please clarify.
5. Please indicate the number of patients above and below the cut-off for inflammatory parameters including CRP and PIV. I think this is very important information for readers. In particular, the cutoff value for CRP was quite high at 17.17, and I thought that the number of patients exceeding the cutoff value may be extremely small. If so, this CRP cutoff value may be statistically useful, but it is not useful in clinical practice. On the other hand, if the number of patients exceeding the cutoff value is not extremely small, it may be useful.
6. For each parameter, please consider whether it is truly useful in clinical practice, taking into account the number of patients exceeding the cutoff value. I think it will enrich the Discussion section.
7. I thought it would be easier to understand if the words tumor size and dimension were unified.
Author Response
Thank you for your constructive and insightful comments.
We have made the following changes and additions based on your suggestions.
- If possible, it may be better to indicate what PIV (pan-immune-inflammation value) is in the abstract. I think it is desirable that the abstract be self-contained.
We have added “Among these biomarkers, the recently identified PIV is a novel biomarker that reflects the immune response and systemic inflammatory status of patients.” to the abstract.
- In the M&M section, the authors stated that they excluded patients with infections and chronic inflammation. Were there any criteria for exclusion, and if not, who determined this? Because the cut-off value of CRP was high at 17.17, I suspect that patients with infection might be included.
In the course of our study, patients with a known history of active infection or chronic inflammatory disease were identified and excluded through the use of retrospective screening of their medical records. This evaluation was based on a synthesis of the clinical notes, laboratory findings and radiological examinations, if available, at the time of diagnosis. Nevertheless, definitive exclusion of infection or chronic inflammation may be constrained by the retrospective nature of the study.
We accept that it was not possible to completely exclude the possibility of active infection among our patients. This is addressed in the limitations section of our study.
- Were there any cases where the patient had multiple blood test in the month before starting treatment? If so, which blood test results were accepted?
Thank you for drawing attention to this important point.
In the present study, only the results of the most recent blood tests were used to assess the pretreatment blood parameters. This approach is predicated on the assumption that blood parameters most accurately reflect the pre-treatment state.
We have added “The most recent results were used if the patient had multiple blood tests within one month of treatment.” to the materials and methods section.
It is acknowledged that this method carries the risk of not completely excluding cases in which blood parameters are unstable. Nevertheless, given the retrospective nature of the study, we believe that this approach offers a standardized and feasible method of analysis. It should be noted that this has been included as a limitation in the discussion section.
We have added “In the case of patients with more than one blood test result within one month before treatment, the result closest to the treatment date was used as the basis for the analysis. It is also possible that this method may not entirely exclude cases in which blood parameters are not stable, which represents another potential limitation of our study.”
- Was FIGO 2018? Please clarify.
Thank you for clarifying this apparently fundamental but nevertheless crucial detail.
We have added “The inclusion criteria are being aged ≥18, having a diagnosis of cervical cancer confirmed by histopathology, being staged as IA-IVA according to the 2018 FIGO staging system, and having serum laboratory results obtained from our hospital's blood analyzer within one month prior to treatment. All included patients have been assigned FIGO staging according to the 2018 FIGO staging system.” to the materials and methods section.
- Please indicate the number of patients above and below the cut-off for inflammatory parameters including CRP and PIV. I think this is very important information for readers. In particular, the cutoff value for CRP was quite high at 17.17, and I thought that the number of patients exceeding the cutoff value may be extremely small. If so, this CRP cutoff value may be statistically useful, but it is not useful in clinical practice. On the other hand, if the number of patients exceeding the cutoff value is not extremely small, it may be useful.
In Table 2, we have included a list of the number of patients whose inflammatory parameters exhibited values above and below the established cut-off points.
And we also have added “The study's results indicated that although the initial cut-off value of CRP was relatively high, 32.2% of patients remained above this cut-off. This finding suggests that the cut-off value does not apply to only a small minority of patients. Therefore, CRP may be considered a clinically useful prognostic marker.” to the discussion section.
- For each parameter, please consider whether it is truly useful in clinical practice, taking into account the number of patients exceeding the cutoff value. I think it will enrich the Discussion section.
We have added “Similarly, the PIV, MLR, PLR, SII, and albumin cut-off values represent approximately one-third of the patient group and have the potential to be useful in clinical practice as a general measure of systemic inflammation. These ratios support the view that these parameters may be meaningful both statistically and in a clinical context. The high LDH cut-off value was statistically significant; however, its applicability to clinical practice may be limited. The high LDH cut-off value resulted in a smaller group of patients (15.6%) being included in this range, which could potentially affect the generalizability of our results.” to the discussion section.
- I thought it would be easier to understand if the words tumor size and dimension were unified.
The term "dimension" was replaced with the word "size."
Round 2
Reviewer 1 Report
Comments and Suggestions for Authors
After first round of revieweng, You were able to address all the issue and rise the quality to the Journal' standards
Author Response
Thank you for your comments.
Reviewer 3 Report
Comments and Suggestions for Authors
Thank you for responding appropriately to all my questions. I feel that these revisions have improved the quality of the paper enough for publication.
Author Response
Thank you for your comments.